# Leprosy and lymphatic filariasis-related disability and psychosocial burden in northern Mozambique

Robin van Wijk [1,2]*, Litos Raimundo[3], Domingos Nicala[3], Yuki Stakteas[3], Adelaide Cumbane[4], Humberto Muquingue[4], Julie Cliff[4], Wim van Brakel[1], Artur Manuel Muloliwa[5]

1 NLR | until No Leprosy Remains, Amsterdam, The Netherlands, 2 Erasmus MC, University Medical Center Rotterdam, Rotterdam, The Netherlands, 3 NLR Mozambique, Maputo/Nampula, Mozambique, 4 Universidade Eduardo Mondlane, Maputo, Mozambique, 5 Universidade Lúrio, Centro de Estudos Interdisciplinares Lúrio, Nampula, Mozambique

* r.vanwijk@nlrinternational.org

**Data Availability Statement:** The data that support the findings of this study are available via: https://

## Abstract

### Introduction

Leprosy and lymphatic filariasis (LF) are among the most disabling neglected tropical diseases (NTDs) that affect the citizens of Mozambique, especially in the Northern provinces. The irreversible impairments caused by these NTDs often lead to psychosocial consequences, including poor mental wellbeing, stigma and reduced social participation. Limited data on these consequences are available for Mozambique, which are urgently needed to better understand the true disease burden and support advocacy for scaling up interventions.

### Methods

A cross-sectional mixed-methods study was conducted. Mental distress was assessed with the Self Reporting Questionnaire (SRQ-20), participation restriction was assessed with the Participation Scale Short (PSS) and perceived stigma was assessed with the Explanatory Model Interview Catalogue affected persons stigma scale (EMIC-AP). Additionally, semi-structured interviews were conducted with persons affected by leprosy or LF.

### Results

In total, 127 persons affected by leprosy and 184 persons affected by LF were included in the quantitative portion of the study. For the qualitative portion, eight semi-structured interviews were conducted. In both disease groups, mental distress was found in 70% of participants. Moreover, 80% of persons affected by leprosy and 90% of persons affected by LF perceived stigma. Moderate to extreme participation restriction was found in approximately 43% of persons affected by leprosy and in 26% of the persons affected by LF. Persons affected by leprosy and LF felt excluded from society and experienced financial problems. More severe disabilities were associated with more severe outcomes for mental wellbeing,

www.infontd.org/sites/default/files/2024-06/240624%20Accessible%20data%20quant%20%2B%20qual%20-%20publication%20InfoNTD.pdf.

**Funding:** *This work received financial support from the Coalition for Operational Research on Neglected Tropical Diseases (COR-NTD), which is funded at The Task Force for Global Health primarily by the Bill & Melinda Gates Foundation, by the UK aid from the British government, and by the United States Agency for International Development through its Neglected Tropical Diseases Program.* Grant number: NTD-SC #165D DATED January 1, 2019. P.O #2707. Funds were awarded to WvB. The funders had no role in study design, data collection and analysis, decision to publish, or preparation of the manuscript.

**Competing interests:** The authors have declared that no competing interests exist.

participation restriction and stigma. By contrast, participation in a self-care group was suggested to have a positive impact on these outcomes.

## Conclusion

The findings provide evidence that persons affected by leprosy and LF must not only confront physical impairments but also experience significant disability in the psychosocial domain, including mental distress, participation restriction and stigma. These challenges must be urgently addressed by NTD programmes to promote the inclusion and wellbeing of persons affected by NTDs.

## Author summary

Leprosy and lymphatic filariasis (LF) are severe tropical diseases that affect over 1 billion people worldwide, especially in poor regions. In Mozambique, where 11 neglected tropical diseases (NTDs) are common, there is limited information about the impact of leprosy and LF on people's mental and social wellbeing. The current study, which occurred in northern Mozambique, with support from NLR Mozambique and health authorities, aims to understand this impact. The research team examined mental distress, participation restriction and stigma using surveys and interviews.

In total, 127 persons affected by leprosy and 184 persons affected by LF were included in the quantitative portion of the study. In addition, eight semi-structured interviews were conducted. Our findings showed that many participants face mental distress (70%), participation restriction (55%) and health-related stigma (86%), with the severity of leprosy and LF linked to greater challenges. Moreover, our analysis revealed that disability, stigma and marital status played a significant role in the presence or absence of psychosocial challenges. Participation in self-care groups was suggested to have a positive impact on mental and social wellbeing.

Despite some limitations, our study highlights the urgent need for better mental and social support for persons affected by disabling NTDs. It is crucial to address mental health, promote inclusion and adapt services to local needs. To this end, we recommend more research with diverse groups and suggest the integration of mental health care into programmes for NTDs. This study emphasises the importance of looking after the overall wellbeing of those affected by leprosy and LF in Mozambique.

## Introduction

Leprosy and lymphatic filariasis (LF) are part of a diverse group of conditions called neglected tropical diseases (NTDs) [1]. NTDs mostly affect people who live in poor communities and have devastating consequences for health, social and economic outcomes. It is estimated that over 1 billion people globally are affected by an NTD. In addition, the World Health Organization (WHO) estimated that NTDs cost around 200,000 lives and 19 million disability adjusted life years each year [1]. In addition, 40% of persons affected by NTDs live in Africa [2].

Leprosy is one of the NTDs that disproportionately affect the poor [3]. Caused by *Mycobacterium leprae*, it manifests in the skin, eyes and peripheral nerves of the hands and feet and can cause permanent physical disabilities, stigmatisation, discrimination and impaired mental

health, which exacerbates the poverty of those affected [3]. Leprosy can lead to blindness and a loss of sensitivity and muscular function in the hands and feet. The WHO divides leprosy-related impairments into three grades: Grade 0 for no impairments, Grade 1 for loss of sensation in hands or feet, and Grade 2 for visible impairments [4].

Lymphatic filariasis (LF) is an NTD caused by nematode parasites. When they invade the body, they cause increasing difficulties in lymphatic circulation, which results in permanent changes in drained tissues; this in turn often leads to lymphoedema or hydrocele and, in worst the cases, elephantiasis [5]. LF is considered one of the leading global causes of permanent or long-term disabilities. Lymphoedema and hydrocele severity can be graded as follows: Grade 1 (mostly pitting oedema and spontaneously reversible on elevation), Grade 2 (mostly non-pitting oedema and not spontaneously reversible on elevation) Grade 3 (elephantiasis and gross increase in volume in a Grade 2 lymphoedema, with dermatosclerosis and papillomatous lesions) [6].

Leprosy and LF are among the most disabling NTDs that affect the people of Mozambique. Although data are scarce, it is known that 11 NTDs have been endemic in Mozambique since 1950, and are mostly prevalent in the country's northern provinces [7]. Grau-Pujol et al. summarised the presence of NTDs and found that little data are available on a national level, they also demonstrated that extensive mapping of the incidence and prevalence of NTDs in Mozambique is needed [7]. In 2021, Mozambique reported an incidence of 3,135 new leprosy cases, around 21% of which were classified as Grade 2 disabilities upon diagnosis [8,9]. Data on the incidence of lymphatic filariasis in the country are even more scarce. The WHO reported that 494,085 persons received mass drug administration for LF in 2021 [10]. It is estimated that around 63% of persons affected by LF worldwide are men with hydrocele and that around 38% have lymphoedema in their lower limbs. Moreover, around 90% of persons who develop lymphoedema are left with chronic manifestations [10].

Although some symptoms of leprosy and LF can be relieved through treatment, both NTDs can cause irreversible impairments, especially when diagnosis and treatment are delayed [11]. These impairments often result in social disadvantages and potentially prevent affected persons from participating in key aspects of life, including work, family and social activities [4,12]. Reduced participation is not only caused by physical impairments but also by multiple psychosocial factors, such as stigma, social exclusion and mental distress. These psychosocial factors are associated with a lower quality of life and aggravate the disease itself and its associated consequences [13]. Several studies have highlighted the chronic consequences of leprosy and LF on mental wellbeing and social participation [14–16]. The WHO recognises that persons affected by NTDs, specifically leprosy and LF, are at risk for mental health conditions. Together, the mental health burden and physical consequences of leprosy and LF have a negative health, social and economic impact on the lives of those affected by the diseases [17].

Although the evidence base for the associations between NTDs and disability, mental health, stigma and social exclusion is growing, data appear to originate from a limited number of countries [13,18]. To date, no data are available on the levels of mental distress, participation restriction and stigma among persons affected by leprosy or LF in Mozambique. However, this information is urgently needed to better understand the disease burden in the country and its endemic regions and to support advocacy for scaling up funding and implementation of available disease management, disability prevention and inclusion interventions [19]. Therefore, the aim of this study is to contribute to descriptions of the psychosocial impact of leprosy and LF on affected persons who live in northern Mozambique.

## Methods

### Ethics statement

Ethical approval for this study was obtained on September 12, 2019 from, the Comité Nacional de Bioética para Saúde (CNBS) in Mozambique, registration number: 32/CNBS/2019. Formal consent was obtained from the study participants: Prior to their inclusion, they were informed of the research aims, the voluntary basis for their participation and their rights. Subsequently, they either signed or thumb printed paper-based informed consent forms. The guardians of all participants below 18 years of age (co-)signed consent forms. All collected data were kept confidential and used for study purposes only.

### Study design and setting

The current study employed a community-based cross-sectional and mixed-methods methodology to better understand levels of physical disability, mental distress, participation restriction and perceived stigma among persons affected by leprosy or LF in Mozambique. The study was conducted in the northern province of Nampula, more specifically in the rural districts of Erati and Memba. Leprosy and LF are endemic in these areas. In these districts, diagnosed leprosy cases are mostly carefully registered at the health centres. In contrast, LF cases are inconsistently registered by local authorities due to the lack of a rigorous surveillance system, these cases are mostly reported by local self-care groups. The study was facilitated by NLR Mozambique and supported by the provincial health authorities. It focuses on the districts of Erati and Memba because these have been reported to have shortcomings in the availability of health and social services. Members of the local community have spoken of high NTD prevalence and a lack of action by local authorities. Data collection occurred from October 2020 to January 2021. The research was undertaken as part of a funded project of the Coalition for Operation Research on Neglected Tropical Diseases (COR-NTD) *Avaliação dos Serviços de Gestão da Morbidade, Prevenção da Deficiência e Inclusão para Pessoas Afetadas pela Lepra, Filaríase Linfática e Konzo.*

### Quantitative data collection

**Sampling strategy and participants.** To determine the sample size for the quantitative portion of the study, a conservative disability and psychosocial burden prevalence of 50% was assumed among persons affected by leprosy and LF since actual data were unavailable. A lower or higher estimated burden would lead to a lower sample size. For the sample size calculation, the statistical programme EpiCalc2000 was used [20]. The parameters were a prevalence of 50% and a precision +/- 10% for the 95% confidence interval. This yielded a sample size of 96. Based on this result, we set a desired sample size of 100 participants per disease type.

Prospective study participants consisted of persons affected by leprosy or LF, including persons with an active infection, persons currently undergoing treatment, and persons who have already completed treatment but still live with negative consequences of the disease, such as impairment, chronic health problems, stigma, psychosocial consequences and/or financial consequences. Potential participants were approached by community volunteers who provide basic health and clinical services, and disseminate information as part of the community health education that they routinely provide. Each volunteer gathered data in their own community and were therefore knowledgeable about where and how to recruit participants. They visited them at their homes to conduct interviews, using the questionnaires. Through convenience sampling (based on the community volunteers' knowledge) and snowball sampling participants were selected for the study. Persons were included if they [1] were affected by leprosy or

LF, [2] lived in one of the target districts (Erati or Memba) and [3] consented to participate in the study. However, persons were excluded from the study if they [1] were affected by more than one of the target diseases (leprosy **and** LF), [2] diagnosed with another disabling condition or [3] were unable to understand the study, its advantages or its risks.

**Data collection.**    The full data collection was conducted electronically on tablets using a mobile application for the data collection software REDCap [21]. The data collection included socio-demographic, health services and disability data.

Three standardised questionnaires were used to assess participants' levels of mental distress, participation restriction and experienced stigma: respectively, the Self Reporting Questionnaire, which included 20 questions (SRQ-20); the Participation Scale Short (PSS), which included 13 questions; and the Explanatory Model Interview Catalogue affected persons stigma scale (EMIC-AP), which included 15 questions [22–24]. The community volunteers received a two-day training, which included disability and lymphoedema grading, conducting the interviews using questionnaires and entering the data into a mobile application via tablets.

First, the questionnaires were forward translated from English to the local language, Makhuwa, by two professional translators. Second, they were translated back to English by a third professional translator who did not have any prior knowledge of the questionnaires. Based on the back translation, several alterations were made to the Makhuwa versions of the questionnaire. Third, the questionnaires were piloted by the volunteers who would gather the data in the two districts under study. Based on their findings, two final versions were developed for each questionnaire, one for each district, due to differences in dialects.

## Socio-demographic, health services and disability data

The interviews began with several questions about the participants' socio-demographic details, such as their age, sex and civil status. Then, questions about available health services concerned topics such as distance to the nearest health facility (in terms of time needed to travel), how the participants travelled there and whether they were part of a self-care group. Finally, disease-specific disability data encompassed disability grade and whether the participant needed assistive devices for mobility.

## SRQ-20

The SRQ-20 was developed to measure mental distress and includes the main signs and symptoms of mental health problems [22,25]. The tool was specifically developed for people living in low- and middle-income countries and is easy to use. The tool demonstrated good reliability in Ethiopia, Rwanda and Zambia, settings that are comparable to Mozambique [26–28]. The SRQ-20 is a screening instrument consisting of 20 yes-or-no questions about the physical and psychological symptoms of mental distress. If a symptom was present in the past month, the question is scored a 1, while a score of 0 indicates that the symptom was absent. Therefore, the highest possible score is 20, and a higher total score indicates the presence of mental distress.

## PSS

The PSS quantitatively measures the severity of respondent-perceived restrictions in social participation. The PSS is a 13-item questionnaire based on the terminology and conceptual framework of the WHO International Classification of Functioning [23]. The PSS is a shortened version of the original Participation Scale, which includes 18 items. The questions are based on key domains in the Participation component of the WHO International Classification of Functioning, such as work, family and community involvement. Participants are asked to compare their level of participation to that of a peer and to assess their level of restriction on this matter.

The scale is easy to use, as it includes yes-or-no questions and a response scale of 1 to 5 for each item. The participation score is the sum of ratings of for all items, with a minimum score of 0 (no participation restriction) and a maximum score of 65 (severe participation restriction). The Participation Scale has been proven valid for use in different cultural environments and suitable for use by non-professional interviewers [24].

## EMIC-AP

The EMIC is based on the framework of cultural epidemiology, and was developed to reflect cultural perceptions, beliefs and practices related to illness [29]. The EMIC-AP is derived from the EMIC and measures anticipated, experienced and internalised stigma related to health conditions. It covers certain aspects of life that can be affected by stigma, such as disclosure, pity, shame, avoidance, respect, being made fun of and marriage prospects [29]. The EMIC-AP includes a list of items that probe the participant's stigma perception and experience [30]. The EMIC-AP consists of 15 items, with possible scores of 0 to 3 for each item. Thus, the total scores range from 0 to 45. The International Federation of Anti-Leprosy Associations (ILEP) experts at a stigma research workshop held in Amsterdam in 2010, and the Neglected Tropical Disease NGO Network (NNN) Stigma guides classified the EMIC-AP as a recommended instrument for measuring perceived stigma among persons affected by leprosy [31].

**Qualitative data collection.** Qualitative methods were used to contextualise quantitative findings and add rich detail to the conclusions. The aim was to include persons affected by leprosy or LF and community leaders from each study district, and a representative from a disabled people's organisation (DPO). The DPO is active in several districts of Nampula, including the studied districts. This led to a target sample size of seven participants. Semi-structured interviews were conducted in July and August 2020. They focused on the available healthcare resources and the needs of persons affected by leprosy or LF for Morbidity Management and Disability Prevention (MMDP) and Disease Management and Disability Inclusion (DMDI) services and were based on the Community Health Assessment and Group Evaluation Action Guide by the Centers for Disease Control and Prevention [32].

MMDP is a broad strategy that involves both secondary and tertiary prevention [33]. Secondary prevention includes strategies and activities to identify diseases early in at-risk groups, to ensure quick treatment and prevent adverse effects. Tertiary prevention includes activities to promote independence of the affected person and prevent further worsening of disability. DMDI is similar to MMDP but is a wider, more holistic concept [34]. DMDI also comprises tertiary prevention and includes psychological and socioeconomic support for people with disabling conditions, to ensure that affected persons have equal access to rehabilitation services and opportunities for health, education, and income. During the interviews, follow-up questions were asked based on the experiences shared by the participants and often included aspects of mental wellbeing and social participation. The interviews were conducted in the commonly spoken Makhuwa language and were audio recorded.

Participants were mostly selected through convenience sampling and occasionally through snowball sampling. In each district, contact persons in the governmental health system (district focal points) were informed about the research team's date of arrival. The research team then made appointments with community leaders and affected persons, based on their availability. Participants were eligible for inclusion if they [1] were affected by leprosy or LF, a community leader, or a member of the DPO, [2] were living in one of the target districts (Erati or Memba) and [3] consented to participate in the study. By contrast, persons were excluded from the study if they were unable to understand the study, its advantages or its risks.

## Data analysis

**Quantitative data analysis.**    Before the analysis, data were exported from the REDCap database to Excel and thoroughly reviewed and cleaned. Records were matched to the paper-based informed consent forms and duplications were deleted. Subsequently, the records were anonymised.

Analysis was performed in SPSS v27. Data were analysed by disease group and aggregated for the overall group of participants. Socio-demographic, health services and disability data were analysed using descriptive statistics and expressed as means, medians, frequencies and percentages. For both diseases, data were collected on participants' level of mental distress using the SRQ-20, the level of participation restriction using the PSS, and level of anticipated and experienced stigma using the EMIC-AP. Cut-off points were adopted from previous studies that used the SRQ-20 and PSS to investigate leprosy. For the SRQ-20, a score of greater than or equal to 8 implied probable mental distress [35]. For the PSS, cut-off categories were as follows: 0–6 indicated no significant restriction, 7–13 indicated mild restriction, 14–30 indicated moderate restriction, 31–50 indicated severe restriction and 51–65 indicated extreme restriction [36]. For the EMIC-AP, in line with the approach by Sermrittirong et al., the cut-off point of greater than or equal to 8 was chosen [37]. This was based on the consideration that participants can be said to perceive stigma when they answer 'yes' to at least four questions, 'possibly' to at least eight questions, or a mix of these. Missing values for questionnaire items were imputed by the study participant's average score on the scale. If more than two values were missing for one scale, the participant was excluded from that scale's total score calculations.

To test the internal consistency of the questionnaires, Cronbach's alpha was calculated. In addition, the results of the three scales were compared between several sociodemographic subgroups. For normally distributed data, a two-sample *t*-test was used to test for differences between two groups. To detect significant differences between multiple groups, a one-way analysis of variance (ANOVA) was used and complemented with the post-hoc Tukey's procedure. When data were not normally distributed, a Kruskal-Wallis test was used to test the significance in differences of median scores. This test detects rank-based differences between groups and determines whether these are significant. When a significant difference was detected, the pair-wise Mann-Whitney test was used to determine differences between each pair of groups within the variables.

Moreover, a multivariate analysis was conducted using linear regression models that included mental distress, participation restriction and stigma as dependent variables. The regression models were bootstrapped for non-normally distributed results. As independent variables all demographic variables potentially associated with the dependent variable, with a p-value of less than 0.2 identified through univariate analysis, were included. Dummy variables were created for the categorical outcomes. Variables with p-values of greater than or equal to 0.05 were eliminated one by one until all remaining variables in the model were statistically significant ($p < 0.05$). The non-normally distributed data were bootstrapped as part of the regression procedure when adding these into the multivariate model.

**Qualitative data analysis.**    Audio-recorded data were translated from Makhuwa and transcribed into Portuguese. A unique identifying number was assigned to each participant to ensure anonymity. The data were subsequently coded using Atlas.ti (Version 8). The coding was inductively performed. First, open coding occurred. Then, codes were derived from the data. During the analysis, a coding scheme was developed. Next, findings in interviews that validated or contradicted one another in the interviews were matched to create data triangulation. Lastly, the findings were discussed with the native Makhuwa and Portuguese-speaking

researchers, and an agreement was reached on the reported results, their interpretation and translation.

## Results

In total, 315 participants were included in the quantitative portion of the study. During the data cleaning process, four persons were excluded from the study because they had a double disease or disability burden: three were diagnosed with both of the studied diseases, while one was also diagnosed with polio.

Table 1 shows the demographics of the total research population included in the quantitative portion of the study and for each disease group. In total, 127 persons affected by leprosy and 184 persons affected by LF were included in the analysis. The mean age of participants was 50 years. Of the 127 participants affected by leprosy, 56 (44%) were women. Within this group, the mean age was 49 years. Fifty-three participants (42%) were recruited in Erati district, while 74 participants (58%) were recruited in Memba district. In addition, 52% of persons affected by leprosy were married, and 40% were single. Of the 184 persons affected by LF, 61 were female (33%). Furthermore, the mean age of persons affected by LF was 52 years. Overall, 76 (41%) of them were recruited in Erati, while 108 (59%) were recruited in Memba. Finally, 77% were married, and 17% were single.

In both districts, participants indicated that they need to travel for a long time to visit a health unit; travel time amounted to around one to two hours in Erati and two or more hours in Memba. However, in Erati, 94% of participants indicated that they travelled by foot, while

**Table 1. Characteristics of the study population (quantitative portion of the study).**

| | | Leprosy | | Lymphatic Filariasis | | Total | |
|---|---|---|---|---|---|---|---|
| | | n | % | n | % | n | % |
| **Total** | | 127 | | 184 | | 311 | 100 |
| **Sex** | Male | 70 | 55.1 | 122 | 66.8 | 192 | 61.7 |
| | Female | 56 | 44.1 | 61 | 33.2 | 117 | 37.6 |
| | Unknown | 1 | 0.8 | 1 | 0.5 | 2 | 0.6 |
| **Age (years)** | Mean (SD) | 49.2 (16.8) | | 51.6 (14.3) | | 50.6 (15.4) | |
| | Unknown | 1 | 0.8 | 5 | 2.7 | 6 | 1.9 |
| **District of residence** | Erati | 53 | 41.7 | 76 | 41.3 | 129 | 41.5 |
| | Memba | 74 | 58.3 | 108 | 58.7 | 182 | 58.5 |
| | Unknown | 0 | | 1 | 0.5 | 1 | 0.3 |
| **Marital status** | Single | 51 | 40.2 | 32 | 17.4 | 83 | 26.7 |
| | Married* | 66 | 52.0 | 142 | 77.2 | 208 | 66.9 |
| | Divorced | 3 | 2.4 | 3 | 1.6 | 6 | 1.9 |
| | Widowed | 7 | 5.5 | 7 | 3.8 | 14 | 4.5 |
| **Disability grade** | 0 | | | | | | |
| | 1 | 18 | 14.2 | 71 | 38.6 | N/A | N/A |
| | 2 | 31 | 24.4 | 71 | 38.6 | N/A | N/A |
| | 3 | 78 | 61.4 | 29 | 15.8 | N/A | N/A |
| | Unknown | N/A | N/A | 13 | 7.1 | N/A | N/A |
| **Included in self-care group** | Yes | 18 | 14.2 | 12 | 6.5 | 30 | 9.6 |
| | No | 109 | 85.8 | 172 | 93.5 | 281 | 90.4 |

*Note.*

*Married or co-habitation

39% of participants in Memba also mentioned the option of using a (motor)bike. Nearly 80% of participants recruited in Erati included persons affected with a Grade 2 disability for leprosy, whereas this prevalence was around 50% for participants recruited in Memba. In the LF group, a more similar rate of severe lymphoedema was found among participants in both districts. In Erati, 53% of participants had Grade 2 or 3 lymphoedema; in Memba this prevalence was 57%. Moreover, 19% of participants with leprosy and 15% of participants with LF in Erati were part of self-care groups. In Memba, this percentage was 11% and 1%, respectively.

Qualitative study results were derived from five interviews with persons affected by leprosy or LF, two interviews with community leaders and one interview with a representative from the DPO. In total, two women and six men were interviewed. Their ages ranged from 21 to 84 years old. They were all farmers, who relied on subsistence agriculture, although some of them were impeded by their disabilities.

## Disease morbidity and disability

In the quantitative portion of the study, the majority of participants affected by leprosy had Grade 2 disabilities (61%), 24% had Grade 1 disabilities and 14% did not have any disabilities (Grade 0). Of the persons affected by leprosy, 18 participants (14%) were part of a self-care group.

In the LF group, most participants had Grade 1 lymphoedema (39%) or Grade 2 lymphoedema (39%). In addition, 16% of participants with LF had Grade 3 lymphoedema, while lymphoedema grades for 7% were not recorded. Out of the 122 men affected by LF included in the study, 96 had hydrocele (79%), and seven had both lower limb lymphoedema and hydrocele (6%). Among the persons affected by LF, 7% were part of a self-care group.

During the qualitative interviews, participants with leprosy reported visible disabilities in their hands and legs. Persons affected by leprosy with disabilities in their legs reported pain, tingling, burning sensations, itching and loss of strength. These all led to a decreased walking ability. Meanwhile, both respondents affected by LF said that they experienced severe pain in their legs, which impeded walking. Moreover, they reported periods of inflammation during which the pain and swelling became more severe. An interview participant from Memba, shared that he was not aware of any services available for his LF-related disabilities: '*There are rehabilitation services for those that fall and break their leg, for example, but I have never seen people that have LF who went there.*'

The representative from the DPO explained that people with disabilities in general are regularly excluded from the community. He himself was left by his parents because of a disability in his leg. He mentioned regular name-calling of persons with disabilities. He also said that persons with disabilities are told that they cannot go to school, and when they do, they are scolded for everything that they do wrong. People with disabilities have difficulties getting married and people are divorcing them.

The results of the three standardised questionnaires are shown in Table 2, while results for each sociodemographic sub-group can be found in Tables 3 (leprosy) and 4 (LF).

## Mental distress

Based on the SRQ-20 cut-off value ($\geq 8$), 69% of persons affected by leprosy experienced mental distress. The percentage of 'yes' answers for mental distress (SRQ-20) can be found in Fig 1. Significant differences were found between the three disability grade groups, with a higher mental distress score found for the higher disability grades ($p < 0.001$). Using the same cut-off value, 70% of persons affected by LF were found to experience mental distress. Although significant differences in mental distress scores were found between persons with different

**Table 2. Scores for each questionnaire and by disease sub-group.**

|  |  | Leprosy | Lymphatic Filariasis | Total |
|---|---|---|---|---|
| **SRQ-20** | n | 126 | 182 | 308 |
|  | Mean (95% CI) | 9.89 (9.0–10.8) | 10.0 (9.3–10.7) | 9.97 (9.4–10.5) |
|  | % above cut-off point (≥8) | 68.5% | 70.1% | 69.5% |
| **PSS** | n | 126 | 184 | 310 |
|  | Median (IQR) | 10 (2–32.5) | 85.4 (0–14.8) | 6 (0–20) |
|  | No restriction (0-6) | 43.3% | 55.4% | 43.3% |
|  | Mild (7-13) | 12.6% | 19% | 12.6% |
|  | Moderate (14-30) | 16.5% | 15.2% | 20.5% |
|  | Severe (31-50) | 20.5% | 8.7% | 26% |
|  | Extreme (51-65) | 6.3% | 1.6% | 8% |
| **EMIC-AP** | n | 127 | 184 | 311 |
|  | Mean 95% CI | 20.3 (18.2–22.3) | 20.5 (19.0–22.0) | 20.4 (19.2–21.6) |
|  | % above cut-off point (≥8) | 79.5% | 89.7% | 85.5% |

*Note.* CI = confidence interval; IQR = inter-quartile range

lymphoedema grades, no clear correlation between mental distress and lymphoedema severity could be detected. An overview of questionnaire scores per disability or lymphoedema grade can be found in Fig 2.

## Participation restriction

The internal consistency of the PSS was found to be good for the total sample (α = 0.92), and for each of the disease sub-groups (α = 0.93; LF α = 0.89). Within the sample as a whole, 33%

**Table 3. Leprosy sub-group: SRQ-20, PSS and EMIC-AP results associated with sociodemographic characteristics.**

|  |  | n | SRQ-20 | PSS | EMIC-AP |
|---|---|---|---|---|---|
|  |  |  | Mean (95%CI) | Median (IQR) | Mean (95%CI) |
| **Sex** |  |  | *p = 0.466* | *p = 0.344* | *p = 0.747* |
|  | **Male** | 70 | 9.5 (8.5–10.7) | 10 (0–33.5) | 20.4 (17.8–23.2) |
|  | **Female** | 56 | 10.2 (8.8–11.7) | 11 (3–33) | 19.8 (16.7–23.0) |
| **Age** |  |  | *p = 0.965* | *p = 0.053* | *p = 0.447* |
|  | Correlation coefficient |  | Pearson: -0.123 | Spearman: 0.174 | Pearson: -0.196 |
| **Marital status** |  |  | *p = 0.152* | ***p < 0.001*** | *p = 0.026* |
|  | **Married*** | 66 | 9.9 (8.7–11.0) | 5 (0–13.25) | 21 (18.2–23.8) |
|  | **Unmarried** | 61 | 9.9 (8.5–11.2) | 27 (4–44.5) | 19.4 (16.4–22.4) |
| **Included in self–care group** |  |  | *p = 0.280* | ***p = 0.005*** | *0.182* |
|  | **Yes** | 18 | 11.1 (9.4–12.6) | 3 (0–11.25) | 23.6 (18.5–28.7) |
|  | **No** | 109 | 9.7 (8.7–10.7) | 11.5 (2.55–38.75) | 19.7 (17.4–21.9) |
| **Disability grade** |  |  | ***p < 0.001*** | *p = 0.105* | ***p < 0.001*** |
|  | **Grade 0** | 18 | 5.2 (3.0–7.4) | 6 (2–22.25) | 8.6 (5.7–11.5) |
|  | **Grade 1** | 31 | 8.8 (7.1–10.6) | 7 (3–17.33) | 17.0 (12.9–21.1) |
|  | **Grade 2** | 78 | 11.4 (10.4–12.4) | 15 (1–40) | 24.2 (21.9–26.6) |

*Note.*

*Married or co-habiting; **bold** = significant after Bonferroni correction (p-value < 0.0025); CI = confidence interval; IQR = interquartile range.

**Table 4. Lymphatic Filariasis sub-group: SRQ-20, PSS and EMIC-AP results associated with sociodemographic characteristics.**

| | | n | SRQ-20 | PSS | EMIC-AP |
|---|---|---|---|---|---|
| | | | Mean (95%CI) | Median (IQR) | Mean (95%CI) |
| **Sex** | | | *p = 0.005* | *p = 0.012* | *p = 0.682* |
| | **Male** | 122 | 9.3 (8.5–10.6) | 5 (0–12) | 20.3 (18.4–22.1) |
| | **Female** | 61 | 11.3 (10.3–12.4) | 9 (2.5–19.5) | 20.9 (18.2–23.6) |
| **Age** | | | *p = 0.345* | ***p = 0.015*** | *p = 0.631* |
| | Correlation coefficient | | Pearson: 0.071 | Spearman: 0.182 | Pearson: -0.036 |
| **Marital status** | | | *p = 0.636* | ***p < 0.001*** | *p = 0.703* |
| | **Married*** | 142 | 10.1 (9.4–10.8) | 5 (0–11) | 20.6 (19.1–22.3) |
| | **Unmarried** | 42 | 9.7 (8.1–11.4) | 11.5 (3.8–36.5) | 20.0 (16.2–23.8) |
| **Included in self–care group** | | | *p = 0.044* | *p = 0.374* | ***p = 0.004*** |
| | **Yes** | 12 | 12.6 (10.9–14.2) | 9 (2.75–21.5) | 28.7 (22.9–32.4) |
| | **No** | 172 | 9.8 (9.1–10.5) | 5 (0–14.75) | 20.0 (18.4–21.5) |
| **Lymphoedema grade** | | | ***p < 0.001*** | ***p = 0.002*** | ***p = 0.007*** |
| | **Grade 1** | 71 | 11.8 (11.0–12.7) | 4 (0–12) | 23.9 (21.5–26.3) |
| | **Grade 2** | 71 | 8.6 (7.6–9.8) | 5 (0–9) | 18.7 (16.4–21.0) |
| | **Grade 3** | 29 | 9.6 (8.0–11.1) | 15 (9.5–30) | 21.1 (17.8–24.5) |

*Note.*

*Married or co-habiting; **bold** = significant after Bonferroni correction (p-values <0.0025); CI = confidence interval; IQR = interquartile range

of participants experienced moderate or worse participation restriction. The percentage of answers confirming participation restriction (PSS) is shown in Fig 3.

Based on the PSS cut-off values (0–6 no significant restriction; 7–13 mild restriction; 14–30 moderate restriction; 31–50 severe restriction; 51–65 extreme restriction), 43% of persons affected by leprosy experienced moderate or worse participation restriction. A significant difference in participation restriction was found between married persons with leprosy (median = 5; IQR = 0–13.25) and single persons with leprosy (median = 20.5; IQR = 2.75–40; p = 0.002). Persons affected by leprosy most frequently reported restrictions on Questions 1 and 2 (work participation): 'Do you have equal opportunity as your peers to find work?' and 'Do you work as hard as you peers do?'.

During the qualitative interviews, a participant affected by leprosy in Memba explained that the availability of assistive devices could help him become more mobile: '*It would be good if there was something that could help me move, maybe a bicycle, then I could reach the health unit and go home more quickly.*'

Similarly, 26% of persons affected by LF experienced moderate or worse participation restriction. Persons with Grade 3 lymphoedema experienced significantly worse participation restriction (median = 15; IQR = 9.5–30) than persons with lower grades of lymphoedema. Persons affected by LF also reported the highest level of restriction on question 1 and 2 of the questionnaire.

During the qualitative interviews, both of the respondents who were affected by LF reported periods of inflammation during which the pain and swelling became more severe. During these periods they had no other option than to stay home and depend on others for water and food, as they were unable to work. One respondent who lived in Erati said, '...*I can't work, I'm a man who should not be working anymore, but I cannot afford to stop ... Money is a real problem ... To have food or not to have food.*'

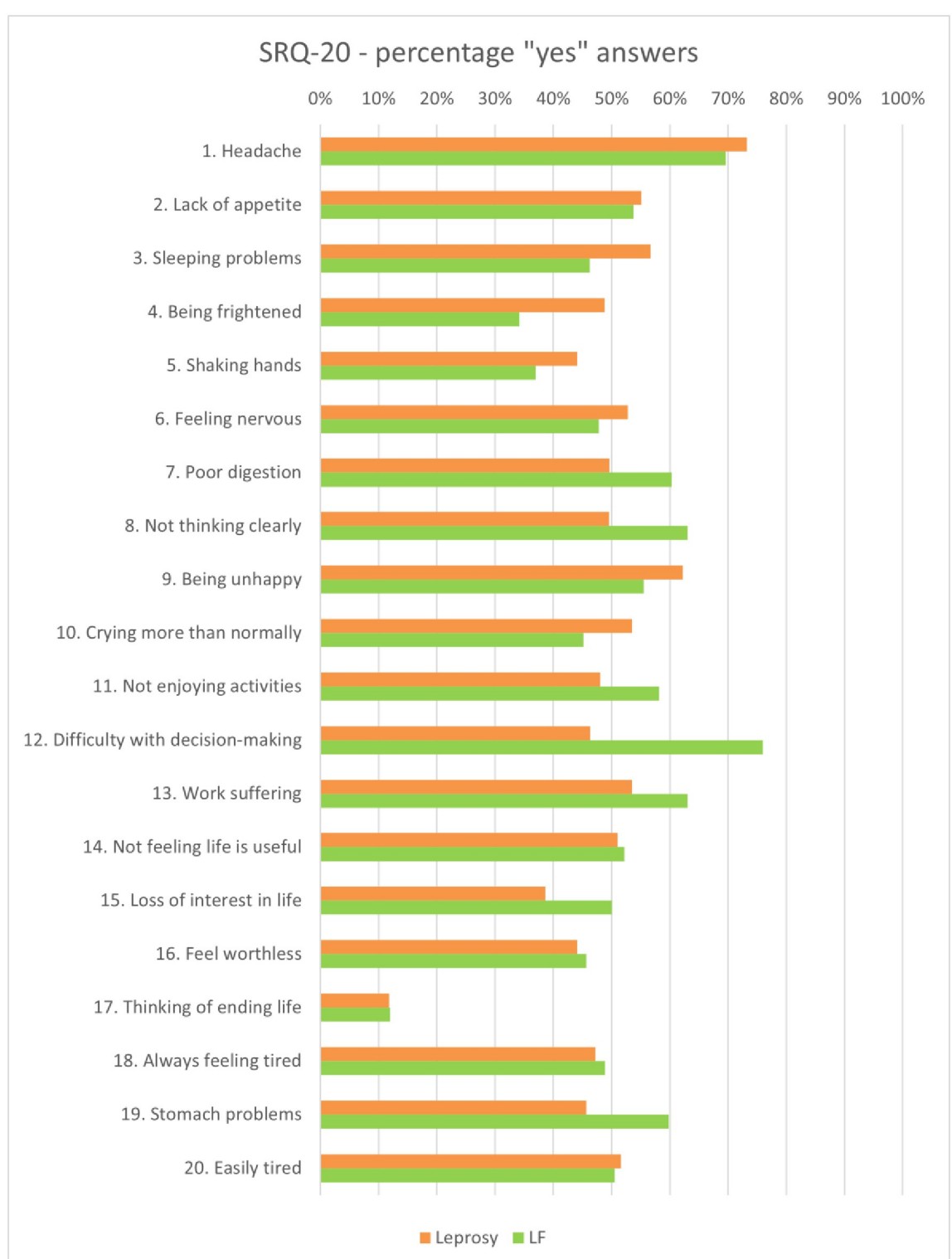

**Fig 1. Percentage of 'yes' answers on the SRQ-20 per disease sub-group.**

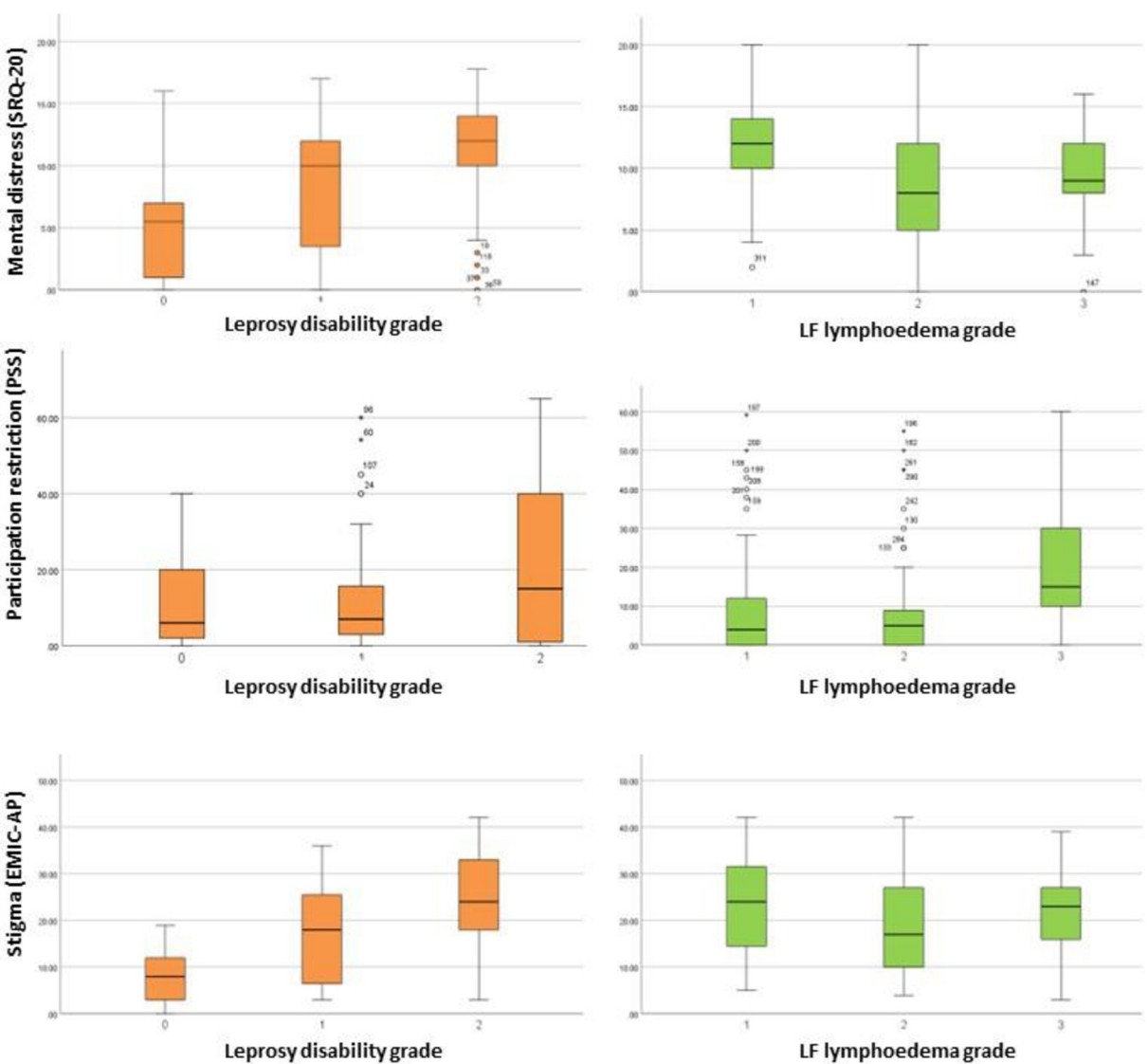

**Fig 2. Mental distress, participation restriction and stigma scores per disability or lymphoedema grade.**

### Stigma

The internal consistency of the EMIC-AP was found to be good for the total population ($\alpha$ = 0.82), and for each of the disease groups (leprosy: $\alpha$ = 0.84, LF $\alpha$ = 0.81). Within the sample as a whole, 86% of participants experienced health-related stigma. The percentage of affirmative answers for stigma (EMIC-AP) is shown in Fig 4.

During the qualitative interviews, one of the community leaders mentioned that he thought that leprosy was caused by a spell and that he was not in favour of letting persons affected by leprosy travel to health units because they may infect many others on their way there. His beliefs had been reinforced by health services' organisation of groups for persons affected by leprosy in the community; he thought that these were a way to prevent affected persons from travelling to health units.

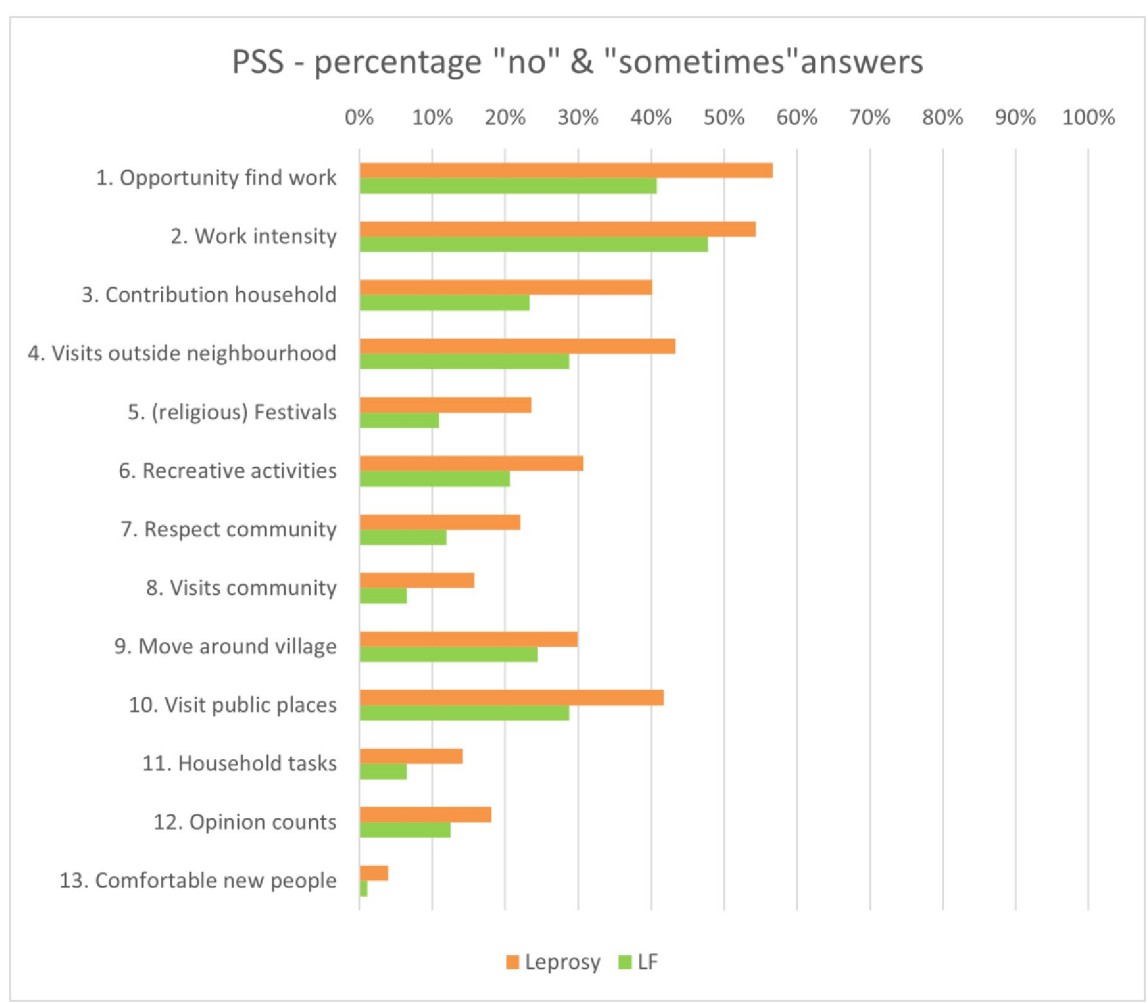

**Fig 3. Percentage of answers that confirmed participation restriction per disease sub-group.**

### Leprosy

Scores on the EMIC-AP scale showed that 80% of persons affected by leprosy experienced health-related stigma. Within this group, a significant difference in experienced health-related stigma was found between persons affected with Grade 2 disability and those with Grade 0 or Grade 1 disability ($p < 0.001$). The difference in EMIC-AP scores between participants with Grade 0 and Grade 1 disability was also significant ($p = 0.016$).

During the qualitative interviews, a person affected by leprosy mentioned that he was told to stay at home and that he needed to be treated with traditional medicine. Another was told that leprosy was incurable and that he needed to be isolated outside of the community. He also explained that he should not sleep next to others or touch others because they may also contract the disease. Moreover, he explained that he should regularly wash because the smell of his sweat could contaminate others. One of the other interviewed persons affected by leprosy said that some people are born with the disease and that they are infected in their mother's womb.

Exclusion was mostly mentioned in the context of leprosy. Community leaders and the representative from the DPO specifically mentioned that persons affected by leprosy were

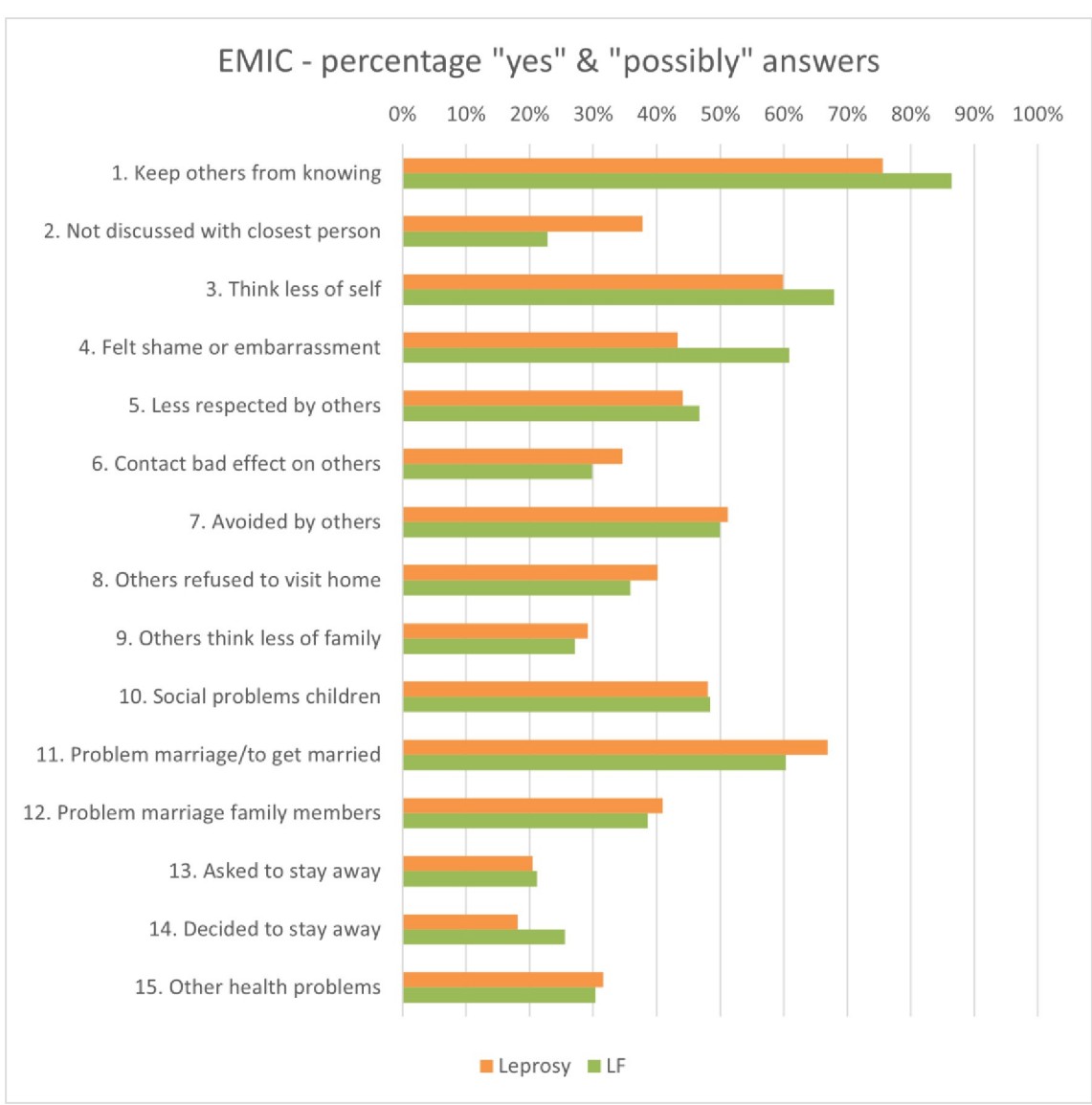

**Fig 4. EMIC-AP Percentage of affirmative answers indicating stigma per disease sub-group.**

regularly excluded from the community due to fear of transmission. They had heard from people affected by leprosy who lived apart from the rest of the community and lacked support from family and friends, which persons affected by leprosy confirmed. One of them explained that he no longer could live with his parents: '*My mother and father despised me. . . I built my own house, where I sleep alone. In this house, I sleep alone and I keep it closed.*'

## Lymphatic filariasis

According to the outcomes of the EMIC-AP questionnaire, 90% of persons affected by LF experienced health-related stigma. In addition, a significant difference was found between persons with Grade 1 and Grade 2 lymphoedema ($p = 0.009$). More specifically, persons with Grade 1 lymphoedema experienced a higher level of health-related stigma than those with

**Table 5.  Multivariate analysis linear regression models for mental distress, participation restriction and stigma.**

|  | SRQ-20 | | | PSS | | | EMIC-AP | | |
|---|---|---|---|---|---|---|---|---|---|
|  | Model | B (p-value) | $R^2$ | Model | B (p-value) | $R^2$ | Model | B (p-value) | $R^2$ |
| **Leprosy** | Disability (yes/no) | 1.728 (0.09) | 0.475 | Married (yes/no) | 13.62 (0.001) | 0.219 | Age | -0.080 (0.070) | 0.514 |
|  | Stigma (EMIC-AP) | 0.271 (< 0.001) |  | Disability (no-mild/severe) | 7.981 (0.016) |  | Disability (no-mild/severe) | 5.121 (0.002) |  |
|  |  |  |  | Self-care group | 9.467 (0.004) |  | Mental distress (SRQ-20) | 1.342 (< 0.001) |  |
| **LF** | Sex | 1.669 (0.004) | 0.362 | Age | 0.251 (0.001) | 0.462 | Self-care group | -5.389 (0.039) | 0.314 |
|  | Lymphoedema (mild/severe) | 1.652 (0.003) |  | Married (yes/no) | 10.42 (0.002) |  |  |  |  |
|  | Stigma (EMIC-AP) | 0.226 (0.001) |  | Stigma (EMIC-AP) | 0.245 (0.015) |  | Mental distress (SRQ-20) | 1.190 (< 0.001) |  |

Grade 2 lymphoedema. Other than these significant differences, no correlation was found between experienced stigma and lymphoedema severity in our sample. Persons affected by LF reported the highest scores on Item 1 ('If possible, would you prefer to keep people from knowing about your disease? ') and Item 3 ('Do you think less of yourself because of your disease? ').

During the qualitative interviews, a man affected by LF from Erati explained that his family members viewed him as an old man who could not do anything after he experienced an attack of inflammation. '...*the whole family knows that "our old man here is sick*".'

## Multivariate analysis

The multivariate models are displayed in Table 5. The SRQ-20 results show that, for persons affected by leprosy, having a disability and experiencing stigma remained significant in the multivariate analysis and accounted for 48% of the variability in mental distress. For LF, 36% of the variability was explained by the female sex, a severe lymphoedema grade, and stigma.

For persons affected by leprosy, 22% of the variability in participation restriction was explained by marital status, with unmarried persons experiencing more severe restrictions; a more severe disability grade; and non-participation in a self-care group. For persons affected by LF 46% of the variability in participation restriction could be explained by age, marital status and experienced stigma.

For persons affected by leprosy, higher levels of stigma were found for younger people, those with more severe disabilities and those with higher mental distress, which explained 52% of the variability in stigma scores. For persons with LF, only participation in a self-care group and higher mental distress remained significant in the multivariate model.

## Discussion

The findings provide evidence that persons affected by leprosy and LF are not only confronted by physical impairments but also experience disability in the psychosocial domain, including mental distress, participation restriction and stigma. This supports the evidence that individual characteristics and environmental factors interact to produce or ameliorate dimensions of disability [38,39]. The questionnaires used exhibited high internal consistency for both disease groups, which indicates that answers were consistent across questions and thus likely to measure the same concept or construct. Mental distress was evident in a substantial proportion of the study population, while participation restriction was experienced by half of the study

sample for both persons affected by leprosy and LF. A high level of perceived and experienced stigma was found in both disease groups and among two-thirds of participants. The existence of stigma towards persons affected by leprosy and LF was widely acknowledged and emphasised during the interviews.

The current findings show that higher disability grades or lymphoedema or hydrocele severity grades were correlated with higher scores on all three scales. This should be taken into account when interpreting and comparing the results with those from other studies, as well as when planning interventions that aim to reduce mental distress, stigma and participation restriction. Over half of the persons affected by leprosy included in this study had Grade 2 disabilities (61%). In 2019 and 2020 the national average among new cases was just over 19% [40]. A recent study on the leprosy incidence in three districts that neighbour Erati and Memba reported a Grade 2 disability rate of 2–14% at the time of diagnosis [41]. This shows that our sample was strongly biased towards persons with more severe disabilities. This is understandable given the use of convenience and snowball sampling, which is likely to lead to persons with visible signs of leprosy. While this still illustrates the severe impact of disability in the leprosy sub-group, it limits the generalisability of the findings. In addition, among persons affected by LF, 79% of men (53% of the total sample) had hydrocele. Furthermore, 67% of persons affected by LF in the sample were male. The global and expected prevalence of hydrocele among men with LF is 63%, indicating that the current study sample may have a slight under-representation of hydrocele cases [10]. Lower limb lymphoedema may be more difficult to hide and leads to other activity limitations than hydrocele. Therefore, it should be considered that the sample was somewhat biased towards persons with lower limb lymphoedema.

The high levels of mental distress observed in the study sample could consistently be related to the stigma experienced by persons affected by leprosy and LF. For persons affected by leprosy, a clear relationship with disability grade was also found. Adekeye et al. described how disease-related stressors can influence the wellbeing of persons affected by NTDs [42]. This includes their livelihoods and results '*in a heightened sense of anxiety or worry among affected populations*' (*p*.i107). This mechanism could also explain the high mental distress found in our population.

Many persons affected by leprosy and LF reported moderate or severe participation restriction. As expected, those with more severe disabilities were most restricted. It is important to consider that it is not only physical impairment that restricts them from taking part in important roles in life; stigma and mental distress also play an important role. Restrictions were especially experienced in participants' ability to work. Van 't Noordende et al. reported similar experiences among persons affected by leprosy and non-filarial and filarial lymphoedema, stating that these persons especially had problems working in the same way as they did before [43].

Persons affected by leprosy reported high levels of anticipated and experienced stigma, which were worse for those with visible disabilities (Grade 2). Persons affected by leprosy preferred not to disclose their disease status and generally thought less of themselves. Leprosy is associated with high levels of stigma [44]. As in the current study, studies from Nepal and Brazil that used the EMIC-AP questionnaire reported that the highest percentage of affirmative answers were also given to the items about disclosure and self-deprecation [45,46]. A qualitative study conducted by Ebenso et al. in Nigeria to explore the stigmatisation of leprosy described how persons affected by leprosy often feel no other option than to conceal their disease status, which leads to treatment avoidance and increases the severity of disabilities [47]. In the current research, higher perceived stigma scores were associated with more severe disability grades, which indicates that this cycle of stigmatisation may also exist in the study population in Mozambique. Very high levels of perceived stigma were also found among persons affected by LF. Two other studies from Nigeria mentioned that persons affected by LF dropped

out of school, experienced difficulties in their marriage, and suffer loss of income [48,49]. Hofstraat and van Brakel performed a systematic review of stigma and NTDs and reported that '*medium to high levels of stigma were found by the 18 studies that cover stigma related to LF*' (*p.* i56) and leprosy-related stigma was described as 'severe' [13]. In the current study, persons affected by leprosy and LF reported extremely high levels of stigma, with similar mean results on the EMIC-AP stigma scale. However, the percentage above the cut-off point was 10% higher in the LF group than in the leprosy group. To date, very little evidence on stigma and NTDs in Mozambique has been published. Therefore the current study findings cannot be compared to other similar studies pertaining to the Mozambican population. An explanation for the high level of reported stigma could be related to the high percentage of persons with (severe) disabilities in the current sample.

Around 70% of persons affected by leprosy and LF experienced mental distress. They had difficulties with thinking clearly and indicated that their daily work suffered. In addition, more severe disability and lymphoedema grades were related to higher levels of distress. Mental distress was also slightly higher for women affected by LF than for men. A study by Van Dorst et al. from Nepal on the mental wellbeing of persons affected by leprosy showed that 38% experienced poor mental wellbeing and about 50% of participants had thoughts about suicide [50]. These findings were also associated with higher disability grades [46]. Two qualitative studies from the same country reported that mental wellbeing was mostly impacted by a lack of support from family and other social networks [14,51]. Obindo et al., who also focused on persons affected by leprosy, found psychiatric morbidity in around 38% of the study population [52]. Studies have shown the significant impact of LF on mental wellbeing. In Nigeria it was documented that 20% of participants suffered from depression, and that 74% had low self-esteem [52]. Abdulmalik et al. reported that persons affected by LF felt unhappy, sad, cried frequently, felt worried, and had suicidal thoughts [49]. These findings are in line with the results from the current study, in which participants frequently expressed feelings of unhappiness. In addition, Lund et al. showed that difficulties with finding and retaining work is associated with poorer mental wellbeing [53]. However, the SRQ-20 is used to screen for mental distress, which is different from identifying depression. A higher percentage of mental distress does not directly indicate psychiatric illness but shows that mental wellbeing is negatively affected.

In this study, mental distress and stigma appeared to mutually influence each other. In any case, it is unlikely that there is only one pathway or direction of influence. In the literature, an overlap has been reported between signs of mental distress and signs of stigma, such as feelings of guilt, decreased feelings of self-worth, and withdrawal and isolation [54–56]. Experiencing stigma could therefore trigger or worsen mental distress, although how someone feels mentally could also affect how they perceive stigma. Sharaf et al. and Oexle et al. indicated that, even when accounting for the influence of depression, people who experience stigma may still have suicidal thoughts [57,58]. Carpiniello et al. found that suicidal ideation can also stem from a significant decrease in self-esteem and a sense of hopelessness, which can also occur among people who experience stigma [59]. This must be considered when providing mental health support to persons affected by NTDs, as there should also be a focus on experienced stigma.

Although only 14% of persons affected by leprosy and 7% of persons affected by LF participated in a self-care group, the results indicated a possible positive impact of such groups on the psychosocial wellbeing of persons affected by leprosy or LF. Deepak et al. conducted qualitative research among leprosy self-care groups in the same province of Mozambique and reported that socialising, better care for disabilities and wounds, and fighting for one's rights were among the main benefits for participants of these groups [60]. They argued that '*self-care activities can have a fundamental role in the prevention of progressive worsening of leprosy-related impairments*' (*p.*290). This was also reported in a country profile for leprosy in

Mozambique, the report stated that '*the perception is that the level of stigma in the community is decreasing, at least in the areas where self-care groups and DPOs* [disabled peoples organisations] *are active*' (*p*.92; 61). In addition to the physical benefits of involvement in self-care group activities, our results show that the psychosocial benefits for persons affected by NTDs should not be underestimated. Moreover, peer support interventions are frequently cited as beneficial for the psychosocial wellbeing of persons affected by NTDs [62–64]. Lusli et al. demonstrated that their counselling intervention effectively reduced stigma, promoted the rights of persons affected by leprosy, and facilitated social participation of their study participants [62]. In a more recent study, Agarwal et al. provided basic psychological support to a small sample of persons affected by leprosy and LF [64]. This intervention positively impacted mental well-being and reduced experienced stigma of the recipients, while also enhancing the sense of well-being among the peer supporters.

## Limitations

The current study has several limitations. One of these is the sampling of study participants. As the recruitment was mainly done by community volunteers, they were more likely to include persons affected by leprosy and LF who were known to be affected and therefore had visible disabilities. This means that persons with less severe disabilities, lymphoedema or hydrocele or who concealed their condition were likely to be overlooked. This led to a study sample which is not representative of the entire population. However, it still shows the high impact of physical disabilities on mental wellbeing, social participation and stigma. A second limitation concerns the cut-off points chosen for the questionnaire scores. The current study did not include a community sample of healthy controls, which prevented us from setting a locally adjusted cut-off point for the SRQ-20 or PSS. However, compared to other studies that used the SRQ-20 a conservative cut-off point was chosen [26,65]. The PSS has not been used as widely, as it is a simplified version of the original Participation Scale and therefore uses different cut-off values. For the EMIC-AP, no cut-off points could be determined through a healthy control group, as not perceiving or experiencing stigma would lead to a score of zero.

## Conclusion and recommendations

This study aimed to contribute to descriptions of the psychosocial impact of leprosy and LF on the lives of those affected in northern Mozambique. The results show that many of them experience severe psychosocial consequences related to their disease and its associated disabilities. Given the stigma experienced by persons affected by leprosy or LF, and the consequences on their mental wellbeing and social participation, these challenges need to be urgently addressed by NTD programmes to promote the inclusion and wellbeing of persons affected by NTDs. This could be done through DMDI services [34], to ensure that they have opportunities for health, education, and income. Furthermore, follow-up research which includes a representative sample of the study population is recommended to make sure that all persons affected by these diseases are targeted by the relevant interventions, such as peer support counselling. In addition, it is important to explore additional strategies for contextualising mental health care approaches in self-care groups, understanding local models to explain and treat persons with mental distress and to integrate NTD interventions with global mental health best practices.

## Author Contributions

**Conceptualization:** Litos Raimundo, Domingos Nicala, Humberto Muquingue, Julie Cliff, Wim van Brakel.

**Data curation:** Litos Raimundo.

**Formal analysis:** Robin van Wijk, Yuki Stakteas, Artur Manuel Muloliwa.

**Funding acquisition:** Litos Raimundo, Humberto Muquingue, Julie Cliff, Wim van Brakel.

**Investigation:** Robin van Wijk, Yuki Stakteas, Adelaide Cumbane, Julie Cliff, Wim van Brakel, Artur Manuel Muloliwa.

**Methodology:** Robin van Wijk, Litos Raimundo, Humberto Muquingue, Julie Cliff, Wim van Brakel, Artur Manuel Muloliwa.

**Project administration:** Litos Raimundo, Domingos Nicala, Yuki Stakteas, Artur Manuel Muloliwa.

**Resources:** Litos Raimundo, Wim van Brakel.

**Software:** Robin van Wijk, Yuki Stakteas.

**Supervision:** Robin van Wijk, Litos Raimundo, Domingos Nicala, Humberto Muquingue, Julie Cliff, Wim van Brakel, Artur Manuel Muloliwa.

**Validation:** Yuki Stakteas, Adelaide Cumbane, Wim van Brakel, Artur Manuel Muloliwa.

**Visualization:** Robin van Wijk.

**Writing – original draft:** Robin van Wijk.

**Writing – review & editing:** Litos Raimundo, Domingos Nicala, Yuki Stakteas, Adelaide Cumbane, Humberto Muquingue, Julie Cliff, Wim van Brakel, Artur Manuel Muloliwa.

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
