## [Decision Letter · Decision Letter 0]

2 May 2024

Dear Dr Robin van Wijk,

Thank you very much for submitting your manuscript "Leprosy and lymphatic filariasis-related disability and psychosocial burden in Northern Mozambique" for consideration at PLOS Neglected Tropical Diseases. As with all papers reviewed by the journal, your manuscript was reviewed by members of the editorial board and by several independent reviewers. The reviewers appreciated the attention to an important topic. Based on the reviews, we are likely to accept this manuscript for publication, providing that you modify the manuscript according to the review recommendations. 

It is the opinion of the three reviewers and the Guest Editor that this is an important study and that it was well-conducted and reported in the manuscript. There are only minor changes to be made before it can be accepted.

Note that Reviewer #3 has put their comments in a copy of the document which is attached.

The authors do not need to make any changes relating to the comments of Reviewer #3 on lines 117, 149 and 601. Otherwise, all other suggested changes will strengthen the manuscript.

There are no conflicts between the reviewers.

Additional comments:

Abstract:

Line 24: add number of interviews.

Methods:

Data Analysis line 267 Multivariable analysis needs more detail as to what models were used and precise description of the outcomes/ dependent variables.

Results:

Add a sub-heading for qualitative part of the study.

Line 315: Write in full DPO

Line 361. Why do Fig 3 &4 get mentioned before Fig 2. They should be numbered in the order that they are presented in the text. The figs with the individual domains should all be presented first before the combined graph of the composite scores.

Table 6 needs more explanation in the caption.

Sincerely,

Lynne Elson, PhD, MPH

Guest Editor

Jong-Yil Chai

Section Editor

It is the opinion of the three reviewers and the Guest Editor that this is an important study and that it was well-conducted and reported in the manuscript. There are only minor changes to be made before it can be accepted.

Note that Reviewer #3 has put their comments in a copy of the document which is attached.

The authors do not need to make any changes relating to the comments of Reviewer #3 on lines 117, 149 and 601. Otherwise, all other suggested changes will strengthen the manuscript.

There are no conflicts between the reviewers.

Additional comments:

Abstract:

Line 24: add number of interviews.

Methods:

Data Analysis line 267 Multivariable analysis needs more detail as to what models were used and precise description of the outcomes/ dependent variables.

Results:

Add a sub-heading for qualitative part of the study.

Line 315: Write in full DPO

Line 361. Why do Fig 3 &4 get mentioned before Fig 2. They should be numbered in the order that they are presented in the text. The figs with the individual domains should all be presented first before the combined graph of the composite scores.

Table 6 needs more explanation in the caption.

Reviewer's Responses to Questions

**Key Review Criteria Required for Acceptance?**

**Methods**

-Are the objectives of the study clearly articulated with a clear testable hypothesis stated?

-Is the study design appropriate to address the stated objectives?

-Is the population clearly described and appropriate for the hypothesis being tested?

-Is the sample size sufficient to ensure adequate power to address the hypothesis being tested?

-Were correct statistical analysis used to support conclusions?

-Are there concerns about ethical or regulatory requirements being met?

Reviewer #1: it is ok

Reviewer #2: The study's aims were clearly described, with a thorough justification laying out the importance of the topic.

The mixed methods are well applied to the aim of the research and biases considered and where possible avoided. 

I would suggest some rewriting of the sentences around 'irreversible impairments' to make it clear that level of ongoing impairments is what matters, but not all symptoms are irreversible.

Reviewer #3: The objectives were clearly articulated.

Study design was appropriate with well described population and hypothesis stated.

Sampling (which was snowballing) has sufficient sample size.

I am not so adept at statistics but I think analysis was good enough

No concern about any ethical breaches.

**Results**

-Does the analysis presented match the analysis plan?

-Are the results clearly and completely presented?

-Are the figures (Tables, Images) of sufficient quality for clarity?

Reviewer #1: it is ok

Reviewer #2: The analysis plan was presented very clearly and followed through in a structured and appropriately described way in the results. Good mix of presentation in tables and highlighting key findings in narrative text.

Reviewer #3: There is a need to reformat some of the Tables (see Review)

**Conclusions**

-Are the conclusions supported by the data presented?

-Are the limitations of analysis clearly described?

-Do the authors discuss how these data can be helpful to advance our understanding of the topic under study?

-Is public health relevance addressed?

Reviewer #1: it is fine

Reviewer #2: Good insight into the sampling bias (towards people with more severe disability), but the analysis did have the ability to distinguish participants with more, or less, disability so appropriate conclusions could be drawn.

Useful interpretation of results around complexity of links between stigma and mental distress.

Reviewer #3: The conclusions were supported by the presented data.

The limitations were clearly stated .

The authors clearly stated the policy implications of the outcome of the study and the health relevance addressed.

**Editorial and Data Presentation Modifications?**

Reviewer #1: minor revision

Reviewer #2: Very nicely written paper, and I don't have many editorial suggestions. I appreciated the clarity of description, particularly around the measures used.

Note my point about wording around irreversibility above.

Reviewer #3: See the reviewer's comments on the article

**Summary and General Comments**

Reviewer #1: Comments

Title: Leprosy and lymphatic filariasis-related disability and psychosocial burden in Northern Mozambique

The title is really very interesting as the authors tried to address one of the most neglected problems.

Abstract

1. Result- line 29-30: More 30 severe disabilities were associated with higher scores for all questionnaires. You have to specify the term questionnaires, to make it clear for the readers. it has to be replaced with the outcomes you want to indicate. 

Methods 

Study design

1. Say a community based cross-sectional mixed-methods study

Sampling strategy and participants

Persons were included if they 1) were affected by leprosy or LF, 2) were living in one of the target districts (Erati or Memba) and, 3) consented to participate in the study. Persons were excluded from the study if they 1) were affected by more than one of the target diseases (leprosy and LF); 2) were diagnosed with another disabling condition or 3) were unable to understand the study, its advantages or risks.

Questions:

2. What to mean by a person affected by leprosy or LF? You have to give clear operational definition for “a person affected by leprosy or LF”. A person affected by leprosy or LF can be different from a person infected by leprosy or LF. Your outcomes like psychosocial burden is the common problem of infected individuals and their family members and/or primary caregivers. Therefore, I want to be clear with your study population.

3. Why you excluded the study participants if they were affected by more than one of the target diseases (leprosy and LF)? For instance, why you didn’t try to see psychosocial burden among those who are affected by both leprosy and LF compared to those who affected by one of them?

Participants

4. Sample size for qualitative study is not clearly stated.

Data collection

5. Clearly elaborate what type of data you collected by using quantitative survey and using interview respectively.

Socio-demographic, health services and disability data

6. The study variables are not clearly stated. For instance, what to mean by included in a self-care group? What is it, how managed, what they do? 

Results

Disease morbidity and disability

7. Line 326, says, Grade 1 Grade 2. May need edition.

8. What to mean by “7% it was not graded”? Is it to mean 0 grade?

9. Table 6. Why you prefer linear regression analysis for categorical outcomes? Why not binary logistic regression?

Reviewer #2: This paper addresses an important and under-researched area. It uses very appropriate methodology to address the research question at hand, and results are presented in a clear way. 

I have no hesitation in recommending its publication.

Reviewer #3: The manuscript is acceptable and well-scripted.

PLOS authors have the option to publish the peer review history of their article (what does this mean?). If published, this will include your full peer review and any attached files.

Reviewer #1: Yes: Tilahun Abdeta Deke

Reviewer #2: No

Reviewer #3: Yes: Prof Taiwo James Obindo

Figure Files:

Data Requirements:

Reproducibility:

References

---

## [Editor Report · Decision Letter 1]

5 Jul 2024

Dear Ms, van Wijk,

We are pleased to inform you that your manuscript 'Leprosy and lymphatic filariasis-related disability and psychosocial burden in northern Mozambique' has been provisionally accepted for publication in PLOS Neglected Tropical Diseases.

Best regards,

Lynne Elson, PhD, MPH

Guest Editor

Jong-Yil Chai

Section Editor

Thank you for addressing all of the issues raised by the reviewers. The manuscript can now be accepted for publication.

<style type="text/css">p.p1 {margin: 0.0px 0.0px 0.0px 0.0px; line-height: 16.0px; font: 14.0px Arial; color: #323333; -webkit-text-stroke: #323333}span.s1 {font-kerning: none

</style>

---

## [Editor Report · Acceptance letter]

22 Jul 2024

Dear Ms, van Wijk,

We are delighted to inform you that your manuscript, "Leprosy and lymphatic filariasis-related disability and psychosocial burden in northern Mozambique," has been formally accepted for publication in PLOS Neglected Tropical Diseases.

Best regards,

Shaden Kamhawi

co-Editor-in-Chief

Paul Brindley

co-Editor-in-Chief
